# Triaxial Electrospun Nanofiber Membranes for Prolonged Curcumin Release in Dental Applications: Drug Release and Biological Properties

**DOI:** 10.3390/molecules30214241

**Published:** 2025-10-31

**Authors:** Sahranur Tabakoglu, Dorota Kołbuk, Paweł Sajkiewicz

**Affiliations:** Institute of Fundamental Technological Research, Polish Academy of Sciences, Pawinskiego 5B, 02-106 Warsaw, Poland; stabakog@ippt.pan.pl (S.T.); dkolbuk@ippt.pan.pl (D.K.)

**Keywords:** electrospinning, core–shell, triaxial fiber, curcumin, release behavior

## Abstract

Triaxial electrospinning was used to fabricate fiber membranes composed of polycaprolactone (PCL), poly(lactic-co-glycolide) (PLGA), and gelatin (GT), designed as carriers for curcumin (Cur) delivery. Here, synthetic polyesters acted as core and shell layers, while GT formed the middle layer containing Cur at varying concentrations. This paper aimed to demonstrate the effect of a shell layer by rearranging the core and shell layers on the kinetics of model drug delivery. In vitro release results indicated the shell layer considerably affected the release behavior, reducing the initial burst release by up to 28% in triaxial fibers compared to coaxial fibers in PLGA-shell forms. The release kinetics were interpreted using the Gallagher–Corrigan model. The membranes were also evaluated for their morphological properties. PLGA-shell-layered triaxial fibers exhibited pore sizes up to approximately 11 µm, small enough to prevent cell migration, while providing higher permeability. The surface wettability analysis of the developed fibers showed that all forms exhibited hydrophilic properties. Furthermore, the cytocompatibility of the fiber membranes was confirmed with the relative cell viability of over 80%. Triaxial fibers with different shell layers displayed similar release trends, yet fibers with the PLGA shell layer demonstrated more favorable performance, attributed to its layer configuration. These findings suggest that the strategic positioning of polymers in triaxial electrospun membranes could be pivotal in optimizing drug delivery systems.

## 1. Introduction

Electrospun nanofiber membranes present a challenging yet promising system for effective and targeted drug delivery, with applications spanning various medical fields, including dental treatments. In a variety of drug delivery applications, conventional electrospinning methods have already been reported [1,2,3]. In previous studies, limitations have been noted, such as the fact that most drugs and biomolecules might not be compatible with organic solvents used in the electrospinning process, as well as factors such as rapid drug release, hydrophobicity of the used polymers, and poor osteogenesis [4,5,6]. To overcome such limitations, studies have focused on multiaxial electrospinning, such as coaxial and triaxial, allowing for the production of more complicated nanofiber architectures, i.e., the core–shell fiber structures. Moreover, thanks to these techniques, polymers that have different good structural properties, such as mechanical and biodegradation properties, can be used to produce better nanofiber structures, and also, additives such as drugs can be used for functionality [7]. The triaxial technique mentioned above is a fairly new method under investigation. In this method, there is a core polymer, and there are two polymers serving as shell layers surrounding the core. It is a promising method to solve the critical limitations in other techniques, e.g., uniaxial and coaxial, such as lack of sustained and controlled drug release, poor solubility of drugs, problems with loading multiple drugs, insufficient mechanical properties and biodegradation, and inadequate biocompatibility [8,9,10].

The objective of this work was to develop three-component nonwovens by triaxial electrospinning to produce core–shell electrospun membranes with prolonged drug release for use in tissue engineering, particularly in dental applications. Therefore, open porosity and pore size are pivotal factors for developed membranes, providing a bacterial and anti-fibroblast growth barrier deep into the tissue. A porous structure is particularly advantageous as it ensures the permeability of nutrients and essential factors necessary for tissue regeneration processes [11]. Conversely, membranes with smaller pores, not exceeding approximately 15 µm, are essential to provide barrier functionality by preventing the migration of soft tissue cells through the membrane to the defect site [12].

In the literature, there are up-to-date reports on the efficacy of triaxial fibrous materials for various applications including drug delivery vehicles [13], bone repair [14], and wound dressings [15,16]. For example, in a study [14], a three-layered membrane was designed to incorporate three different additives, each intended to support different stages of the bone repair process. Their findings confirmed that the triaxial membrane developed provided a synergistic effect for bone tissue repair by combining biological additives for enhanced cell interaction and antibacterial properties, along with an appropriate material design that ensured controlled degradation and sufficient mechanical support. In another research [17], it was demonstrated that designed triaxial membranes exhibited long-term antimicrobial activity over coaxial and uniaxial fiber membranes, confirming their high potential in various applications. To the best of our knowledge, no report has yet been presented in the literature on a triaxial nanofiber membrane developed for barrier membranes for use in dental tissue regeneration applications. Moreover, while many studies have compared triaxial fibrous materials with their coaxial or uniaxial counterparts, there appears to be a lack of systematic research addressing the effect of different compositions within triaxial fibers on material performance.

In this study, we developed core–shell and triaxial fibers to analyze their potential for use in tissue engineering, especially dental applications. Developed triaxial membranes were compared to their coaxial counterparts as a control group to analyze changes in Cur release as a model drug. Therefore, in both types of core–shell fibers, the pharmaceutical ingredient was inserted into the same polymeric component: the middle layer in triaxial fibers and in the shell layer in coaxial fibers. As a pharmaceutical ingredient, Cur was incorporated into electrospun membranes. Cur is a natural phytochemical polyphenol that possesses antimicrobial [18], antioxidant [19], and anti-inflammatory [20] properties. Cur has also been demonstrated to accelerate the formation of granulation tissue, tissue remodeling, and collagen deposition, all of which contribute to wound healing [21,22]. Furthermore, it has some disadvantages, including low water solubility and low in vivo bioavailability [23]. Cutting-edge electrospinning technology can generate a variety of fibrous structures that facilitate controlled drug release for such challenging water-insoluble bioactive compounds [24].

Here, three different polymers, including PCL, PLGA, and GT, were used to fabricate electrospun membranes. PCL, a member of the aliphatic polyester family, is widely utilized for tissue engineering applications due to its high processability and capability of sustaining its stiffness in physiological ambient conditions [25]. PLGA is a co-polymer consisting of monomers of poly-lactic acid (PLA) and poly-glycolic acid (PGA) [26]. GT, derived from collagen, is a widely utilized natural polymer that increases the biocompatibility of the material [27]. Due to its high hydrophilicity and fast degradation rate, it is preferred in combination with synthetic polymers which have a relatively hydrophobic structure and better mechanical strength, to obtain an optimum material system aimed to be used in tissue engineering applications [28]. We developed two different compositions of electrospun membranes by positioning those polymers in different layer configurations. In Composition 1, PCL was employed as the core layer, GT was positioned as the middle layer, and PLGA was used as the shell layer. In Composition 2, PLGA served as the core layer, GT was maintained as the middle layer, PCL was employed as the shell layer.

This study aimed to investigate the influence of the morphology, architecture, and chemical composition of multicomponent fibers, both with and without an additional shell layer, on the release profile of Cur. In the developed system, Cur was incorporated into the GT layer, which was positioned between Cur-free polyester layers. While the present study focuses on the morphological, drug release, and cytotoxicity aspects of the developed system, a comprehensive structural and physicochemical characterization, including thermal, crystallographic, and spectroscopic analyses will be presented in a complementary study.

## 2. Results and Discussion

### 2.1. Morphology

Electrospinning conditions for plain and Cur-loaded core–shell fibers were identified through preliminary electrospinning trials to produce bead-free fibrous meshes using Composition 1 and Composition 2. Utilizing the final parameters that have been selected (as reported in the experimental section), successful fabrication of defect-free coaxial and triaxial fibers was achieved.

In both compositions, Cur-free and Cur-loaded fibers exhibited bead-free and uniform surface morphology, without any Cur aggregates in all compositions (Figure 1 and Figure 2).

In Composition 1, with a PCL core and PLGA shell layer, the fiber diameter distribution was homogeneous, with a maximum c.a. 1110 nm. Cur-loaded fibers exhibited a larger diameter, which was proportional to the amount of Cur (Figure 1 and Figure 2). It seems that the Cur loading affected the fiber diameter. 

In Composition 2, with a PLGA core and PCL shell layer, it was observed that there was a more heterogeneous distribution in fiber diameters. Furthermore, fibers of Composition 2 are likely to become flattened and fused in some spots. Flattened fibers are known in the literature as ribbon fibers. The observation of such morphology depends on various factors, including solvent evaporation, solution conductivity, polymer concentration, and electrospinning process parameters [29]. Yet, an in-depth analysis of how these structures appear is beyond the scope of this research. It might be further addressed in a separate study.

The data given in Table 1 demonstrate that the composition of the system affects the layer thicknesses in the core–shell-architected fibers. The distribution of the thicknesses is more homogeneous in Composition 1 compared to Composition 2. It might be that there is a higher material interaction between the layers in Composition 2 during the electrospinning, which can lead to partial mixing rather than discrete layers. Furthermore, it could be a consequence of the varying stretching efficacy at the same applied voltage due to the differences in flow rates and viscosity.

Diameters of the middle layer in the triaxial systems in both compositions were determined additionally in fluorescence microscopy, after Rhodamine B (RhB) staining (Figure 3). The diameter obtained with FM is less than the entire diameter seen in SEM pictures for both compositions. FM images prove that the RhB concentration is in the middle layer, as is expected, which indicates the multilayered architecture of the fibers. The shell layer is not visible in FM as it lacks RhB fluorescent dye. Moreover, for the RhB-loaded triaxial fibers, the ratio of the total diameter (as determined from SEM images) to the total diameter obtained by FM indicates that the inner layer (sum of middle and core layers) is comparable to that ratio found for the curcumin-loaded fibers (calculated from the data of Table 1 and Figure 1 and Figure 2) for both compositions. The calculated ratios of Composition 1 are ~1.78 and ~1.81, respectively. And for Composition 2, the ratios are ~2.16 and ~2.27, respectively. Although the diameters obtained by SEM differ between the RhB- and Curcumin-loaded samples, the consistent ratio shows that the layer distribution is similar. And this suggests the reproducibility of fabrication processes across different samples. It is a promising insight from the perspective of taking into consideration different types of pharmacological substances for different purposes.

Porosity is critical in such scaffolds because it directly influences nutrient diffusion, vascular infiltration, and cell migration, all of which are essential for effective tissue regeneration. Additionally, the electrospun membrane, with its smaller pore size, acts as a barrier against undesired fibrous tissue infiltration, ensuring that the regenerative process is confined to the defect site. Proper porosity enables the scaffold to simultaneously protect the defect from soft tissue invasion and promote controlled bone regeneration [30].

Our results indicate pore sizes with a maximum of approximately 11 µm, being sufficiently small to obstruct cell migration in both compositions (Table 2). When comparing the pore sizes of triaxial fiber membranes in both compositions, Composition 2 exhibited smaller pore sizes, with 4.2 ± 2.7 µm for Tri_1% and 5.2 ± 0.7 µm for Tri_3%, compared to Composition 1, which had 10.3 ± 1.1 µm for Tri_1% and 11 ± 0.6 µm for Tri_3%. The pore sizes of Composition 2 might be too small to permit the permeability of essential factors. However, Composition 1 can serve as a barrier membrane, meeting cell occlusivity requirements while offering higher permeability regarding the information provided in the introduction chapter. Furthermore, as Composition 1 offers higher permeability for essential factors, the need for pharmaceutical additives is less, ensuring non-toxicity with the use of smaller amounts.

### 2.2. In Vitro Drug Release Tests and Curcumin Release Kinetics

The cumulative Cur release profiles, as measured by a UV spectrometer (Figure 4) and recorded over a 14-day period, exhibited a biphasic release profile with a high rate in the first 8 h followed by a steady Cur release throughout the period. Many controlled-release systems exhibit an initial burst release, which can be attributed to a variety of processes, including pore diffusion, surface desorption, or the absence of a diffusion front barrier [31].

It is evident from Figure 4A,B that there is a saturation of release for coaxial structures reached between 50 and 75 h, while in the case of triaxial fibers, the high release rate stage observed at the beginning is followed by a continuous very slow release over the whole time of the experiment. The slow release of Cur in triaxial fibers during the late stage provides a promising possibility for chronic disease treatments [32]. Taking a look at the effect of Cur content on the release process, it is clear that the efficiency of the release is better for higher (3%) compared to lower (1%) Cur content. This difference is large for Composition 1 (ca. 10%), with a very small difference, if any, for Composition 2. Such a change in cumulative release in relation to the amount of Cur indicates that the diffusion driving force was the primary factor for controlling the release of Cur from the fibers. Similar results were also reported in previous works with Cur [33,34]. In addition to these, the amount of initial release within 24 h significantly decreased in triaxial fibers over coaxial fibers in Composition 1, as can be seen in the plots. This means the third layer plays an active role as a shell layer, providing a physical barrier to activity. These findings are also confirmed with the data of the mathematical model-fitting in the following part of this section (Table 3).

In addition to these, it was observed that the initial release from the triaxial fibers at the first stage was higher in Composition 1 than in Composition 2. This is because of the amorphous structure of PLGA as a shell layer in Composition 1 and the semicrystalline structure of PCL as a shell layer in Composition 2. Moreover, the release pattern of Composition 2 is not as uniform as in Composition 1. This might be because of the aforementioned reason and also due to the incomplete shell layer closure and, hence, the partial mixing of GT on the surface. This leads to a faster release from that side, resulting in a non-stable profile. These results suggest that the core–shell fiber architecture was properly provided in Composition 1, resulting in a more uniform and sustained release over the experimental period. It is worth noting that SEM images of the meshes regarding the uniformity of surface morphologies and core–shell fiber architecture are in correlation with the release profiles.

The Gallagher–Corrigan mathematical model was employed to investigate the release kinetics and underlying mechanisms of curcumin released from the developed coaxial and triaxial polymeric fibers (Figure 4A,B). The correlation coefficient values (R^2^) (>0.99) indicate that the model is well adjusted to the experimental data. This model characterizes drug release as a biphasic process, comprising an initial burst phase followed by a sustained release phase [35]. The data obtained (Table 3) confirm that curcumin release follows this biphasic mechanism, governed by two distinct rate constants, k_1_ and k_2_. The initial phase (k_1_) corresponds to diffusion-driven drug dissolution into the surrounding medium, while the second phase (k_2_) reflects the slower, degradation-controlled release of curcumin from the polymer matrix [36].

The initial (burst) release rate (k_1_) was consistently lower in all triaxial formulations compared to their coaxial counterparts. For instance, Tri_2_3% exhibited the lowest k_1_ value (0.130), representing a 76.4% reduction relative to Co_2_3% (k_1_ = 0.550). This suppression of the initial release rate is attributed to the additional diffusion barriers and multilayered structure of the triaxial fibers. Similar results were also reported by Jardim et al. [37] for Cur release from layer-by-layer nanoplatforms. Moreover, the fact that k_1_ > k_2_ across all formulations indicates that the diffusion-driven release of Cur occurred at a faster rate than the release of Curcumin through matrix erosion.

Additionally, samples with reduced release rate constants and lower Tm values correspond to fibers with lower porosity and smaller pore size. This can be explained by the fact that limited porosity and smaller pores restrict the penetration of the dissolution medium into the fiber matrix, thereby slowing drug diffusion. Similar results have also been reported in research on porous core–sheath nanofibers fabricated by coaxial electrospinning [38]. In another study on uniaxial fibers, it has been reported that non-porous nanofibers exhibited the highest drug release, followed by porous microfibers and non-porous microfibers [39]. These findings indicate that fiber diameter, and consequently surface area, plays a significant role in the drug release profile alongside surface porosity. While our data indicate that pore size and fiber diameter both influence release kinetics (higher initial rate, but lower cumulative release with smaller pores and a higher fiber diameter as in Tri_1_1%), attributing these outcomes to a single parameter would be an oversimplification for these complex core–shell systems. More detailed structural analyses are required to fully elucidate the underlying mechanisms.

In terms of cumulative release, Tri_2_1% and Tri_2_3% exhibited initial burst releases of ~68% and ~75%, respectively, whereas Tri_1_1% and Tri_1_3% demonstrated lower initial releases of ~53% and ~63%, respectively. Although k_2_ values were relatively similar across all formulations (ranging from 0.043 to 0.225), triaxial systems generally displayed slightly lower or comparable k_2_ values, suggesting a more prolonged second-phase release. Notably, triaxial fibers with Composition 1 exhibited lower k_2_ values than those with Composition 2, further indicating slower release during the degradation-controlled phase.

Such differences in the release profile can also be attributed to the differences in the glass transition temperature (Tg) of the shell materials. In particular, at physiological conditions (around 37 °C), PCL’s very low glass transition temperature (Tg~−60 °C) results in a rubbery state with high chain mobility and free volume, greatly enhancing drug diffusivity through the PCL layer [40]. In contrast, PLGA remains in a glassy state under the same conditions due to its higher Tg (~35–55 °C, depending on co-polymer composition), which reduces the mobility of polymer chains and permeability of the PLGA layer [41,42].

These results demonstrate the effect of the outer shell layer in modulating release behavior among the coaxial and triaxial fiber formulations. Furthermore, the two triaxial compositions differ configurationally, contributing to distinct release kinetics. The variations in kinetic analysis results between Tri_1 and Tri_2 highlight the critical influence of shell layer composition on curcumin release.

Considering the therapeutic objective of long-term Cur delivery, Tri_1 formulations, particularly Tri_1_1%, seem to be suitable candidates. They exhibit significantly reduced burst release, delayed release peaks, and incomplete early Cur release, effectively preserving a portion of the loaded Cur for extended diffusion. In contrast, Tri_2 formulations, which release over 90% of the Cur within the first few days, are less suitable for tissue engineering applications requiring prolonged delivery durations.

### 2.3. Surface Wettability 

The analysis of water contact angle vs. time of contact (Figure 5A,B) indicates that in both compositions, all fibers’ water contact angle drops quickly from a maximum of 72 degrees to 0 in a short period of time. It confirms that all fibers exhibit hydrophilic characteristics, and the differences among them were not statistically significant (*p* > 0.05). Although Cur is not water-soluble, the addition of Cur did not provide a hydrophobic characteristic to the fibers as seen in the water contact angles. This might be because of the low concentrations of Cur. Similar observations have been reported on Cur-loaded PCL/gum tragacanth [43] and PCL/Gelatin [44] electrospun fibers. This observation may also imply that curcumin must be embedded within the polymer matrix rather than being present on the surface, as the latter would likely result in increased water contact angles, especially in coaxial structures. Furthermore, the fact of no alteration in hydrophilicity may even be considered beneficial for cell proliferation and healing at the wound site [45,46].

Moreover, in triaxial fibers, the hydrophobic characteristics of PCL and PLGA were not dominantly observed as shell layers. However, the WCA of Tri_3% in Composition 1 was reduced more slowly than that of the other triaxial fibers in both compositions. The reason might be that the shell layer covered the inner layers much more properly in Tri_3% of Composition 1 compared to the other triaxial fiber samples. In the other samples, there could be some partial gelatin on the surface as a result of the electrospinning process, leading to interaction with water.

In addition to these, no direct correlation was observed between the WCA measurements and the corresponding curcumin release profiles. This indicates that curcumin release is more strongly influenced by morphological parameters and fiber architecture, e.g., shell layer configuration. Moreover, other structural factors may also play a dominant role.

### 2.4. Biological Evaluation

To assess any potential cytotoxicity caused by the polymer matrix or the released Cur, L929 fibroblasts were cultivated in the electrospun mesh extraction medium for 24 h. Cell viability assessment was performed using Cur-free and Cur-loaded meshes, including crosslinked and non-crosslinked samples. Relative cell viability was determined by the Presto Blue assay. Figure 6 show that cells cultured in the extraction medium of fibers of Composition 1 and 2, both without and with Cur, exhibited relative cell viability of over 80% for all samples, including the samples after crosslinking (GTA). It showed that crosslinking had no adverse effect on the cytotoxicity of samples. In the case of Cur, there is a noticeable reduction in cell viability for higher Cur concentrations in coaxial fibers, as reported by other authors [47]. In the case of triaxial fibers loaded with different Cur concentrations, no significant difference (*p* > 0.005) was found between their cell viability results. This suggests that the third layer in triaxial fiber plays an effective role as a shell layer, providing a controlled and slower release. This result is consistent with the Cur release results, confirming the existence of a shell layer. Additionally, slightly higher cell viability in all samples of Composition 2 over Composition 1 may be another confirmation that the core layer is not fully covered and the GT layer is partly on the surface, and this induces cell viability. Previous reports indicated that Cur could exhibit cytotoxicity in a dose-dependent manner, where the cell viability is less than 50% [48,49]. Here, we could confirm that both concentrations of Cur have low cytotoxicity, and this is in line with the earlier study that demonstrated that a gradual release of Cur from nanofibers caused minimal cytotoxicity to fibroblasts [50].

The L929 fibroblast cells, known for their role in producing the extracellular matrix and their adherent nature, exhibited comparable morphology when cultured in electrospun samples, Composition 1 and Composition 2 extracts, and on tissue culture plastic (TCP) (Figure 7 and Figure 8). Cell proliferation in the presence of the sample extracts remained consistent with that observed in the TCP control group. After 24 h of incubation, the fibroblasts retained their characteristic actin cytoskeleton structure, visualized in green, with nuclei stained in blue. Fluorescence microscopy revealed a well-spread cellular appearance, with visible filopodia and lamellipodia, suggesting effective interactions between the cells and the surrounding material.

## 3. Materials and Methods

### 3.1. Materials

PCL (Mn = 80 kDa), GT from porcine skin, type A (gel strength 300), phosphate buffer saline tablets, and Tween 80 were purchased from Sigma-Aldrich, St. Louis, MO, USA. PLGA with PLA/PGA in the ratio of 82:18 was purchased from Corbion (Amsterdam, The Netherlands). Hexafluoroisopropanol (HFIP) (purity degree 98.5%) was procured from Iris Biotech GmbH (Marktredwitz, Germany). Cur was purchased from Apollo Scientific, (Whitefield Rd, Bredbury, UK). Glutaraldehyde (conc. 25%) and sodium azide were purchased from Chempur (Piekary Śląskie, Poland). All chemicals were used as received without further purification.

### 3.2. Electrospinning and Preparation of Samples

In this work, four different types of nonwovens were developed using electrospinning equipment: coaxial and triaxial fibers with and without Cur.

All the polymer samples were dissolved in HFIP at room temperature and subsequently left to stir overnight. For electrospinning, amounts of 5 wt. % of GT and 5 wt. % of PCL as well as 6 wt. % of PLGA 82:18 were prepared. The solutions flowed through a stainless steel triaxial needle spinneret EFA040 (SKE Research Equipment, Bollate, Italy) with the 14 G outer needle, 18 G middle needle, and 24 G inner needle, with flow rates of 1.25, 0.75, and 0.25 mL/h, respectively, for Composition 1 and flow rates of 1.00, 0.75, and 0.25 mL/h, respectively, for Composition 2. Here, two different compositions of triaxial electrospun meshes were developed to assess Cur release performance over their coaxial forms. In the composition called Composition 1, PCL, GT, and PLGA were used as core, middle, and shell layers, respectively. In the other composition called Composition 2, PLGA, GT, and PCL were used as core, middle, and shell layers, respectively. Coaxial counterparts as control groups were developed without the third layer, i.e., without the PLGA shell for Composition 1 and without the PCL shell layer for Composition 2. Thus, the shell (outer) layer in coaxial fibers was a GT layer. All fiber forms were fabricated with and without Cur in the GT layer. The architecture of the fibers developed is illustrated in Figure 9, and the constituents of those architectured compositions are shown in Table 4 and Table 5. The inner layer, intermediate layer, and outermost layer of a whole fiber were labeled as the core layer, middle layer, and shell layer, respectively. For all, the process was carried out at the temperature range of 22–25 °C and humidity range of 30–40%. Other process parameters were as follows: distance between needle and collector, 15 cm; voltage applied to the needle, 12–15 kV. Electrospinning parameters were chosen through preliminary trials and SEM observations (JEOL JSM-6390LV, Tokyo, Japan) to ensure stable jet formation and uniform, bead-free fibers.

For Cur-loaded samples, the same methodology of electrospinning was applied. Cur was dissolved in HFIP first and then added to the electrospinning solution of GT to prepare a blended solution before the electrospinning. After electrospinning, all prepared meshes were vacuum-dried to evaporate the solvent residuals, and then crosslinking was performed. Briefly, a 25% glutaraldehyde solution was transferred to the bottom of a desiccator covered by a lattice-shaped plate with the developed meshes on top. The desiccator was sealed for 2 h at ambient conditions; then, the crosslinked samples were vacuum-dried to remove the solvent residues. 

For morphological analysis under fluorescence microscopy (Leica AM TIRF MC, Wetzlar, Germany), electrospun fibers with Rhodamine B (RhB) in the gelatin layer were produced. RhB was loaded at 0.1% wt. into gelatin, and the same production conditions mentioned above were applied to produce electrospun fibers of Composition 1 and Composition 2. Fibers developed were analyzed under fluorescence microscopy and scanning electron microscopy for comparison. 

### 3.3. Characterization

#### 3.3.1. Morphology of Fibers

The morphology of the developed nonwovens was analyzed using a scanning electron microscope SEM (JEOL JSM-6390LV, Tokyo, Japan). All samples studied were first coated with a gold layer of 10 nm thickness using a sputter coater (Smart Coater DII-29030SCTR, Osaka, Japan). During the scanning process, the acceleration voltage was in the range of 7–10 kV and the working distance was 10 mm. The nonwovens’ SEM images were used to measure the fibers’ diameters with ImageJ software (v. 1.51j8, Bethesda, MD, USA), where the measurement method involved the intersection of drawn lines with the fibers. Fluorescent images of the triaxial fibers were recorded with fluorescence microscopy (Leica AM TIRF MC, Wetzlar, Germany), which aimed to analyze the distribution and alignment of layers in the fibers.

##### Estimation of Inner Layer Thicknesses

Utilizing the overall fiber diameter obtained through SEM imaging, the inner layer diameters of the coaxial and triaxial core–shell electrospun fibers were determined. The methodology was developed using the model for coaxial fibers [51]. Some critical assumptions were employed to apply the approach: (1)The shell (outermost) layer has a uniform thickness around the fiber. (2) The solvent system used for electrospinning is consistent across all fibers, leading to a similar distribution and density of the polymer solution. (3) The total fiber diameter consists of the diameters of the core, middle, and shell layers (Figure 10). Considering these, it was assumed that the diameters of the inner layers can be derived as fractions of the total diameter based on their proportion in the coaxial/triaxial structure. This proportionality is determined by the relative concentrations, flow rates, and volume of the needle where the solution flowsthrough, as used during electrospinning. Assuming that each layer has a uniform thickness around the fiber axis, the substitution of the diameters between each layer gives the thickness for that layer.

The volume of chosen fiber length might be correlated to the theoretical area of each layer (*S*) because the lengths of each layer are the same. Additionally, due to the known fiber diameter from SEM related to the shell layer, the relation of the theoretical area to the flow rate of the solution may be calculated from the following equation (Equation (1)): (1)$$C=\frac{S}{Q}=\frac{\pi r^{2}}{Q}$$
where *C* is the constant obtained by the ratio of the theoretical area of the layer (*S*) to the flow rate of the solution for the same layer (*Q*), and *r* = *D*/2.

Assuming a similar distribution and viscosity of solutions used in the system, the relation of fiber diameter to the constant *C* for each layer is the same (Equation (2)):(2)$$\frac{D_{shell}}{C_{shell}}=\frac{D_{middle}}{C_{middle}}=\frac{D_{core}}{C_{core}}$$
where the *D_shell_*, *D_middle_*, *D_core_* are the diameters of each layer of the fiber, and *C* is the constant obtained by Equation (1). *D_shell_* is a known parameter obtained from the SEM images of the fibers as the fiber diameter. Hence, the *D_middle_* and *D_core_* were estimated by applying Equation (2).

##### Porosity

The average porosity, *p*, of the electrospun fiber meshes was estimated by applying the following equation (Equation (3)) [52]:(3)$$p=\frac{V_{t}-V_{f}}{V_{t}}=1-(V_{f}\times (\frac{d}{m_{t}}))=1-(\frac{m_{f}}{m_{t}})$$
where *m_t_* is the theoretical mass of solid specimen calculated as a function of volume occupied by a patch *V_t_*, and the density of material *d* (Coaxial fibers; ~1.25 g/cm^3^, Triaxial: ~1.19 g/cm^3^), which are estimated considering the densities of the PCL (1.14 g/cm^3^), PLGA (1.15 g/cm^3^), and GT (1.3 g/cm^3^) and their percentages of total mass in the specimen; *V_t_* and *m_f_* are the real specimen volume and mass, respectively. Recent studies have proposed alternative versions of this equation while maintaining the same underlying principles [53,54,55]. These updated formulations have been applied to a variety of materials, including uniaxial fibers and coaxial fibers, as well as hybrid materials, thereby supporting the broad applicability of the method.

Furthermore, the average pore size (*P_c_*) was determined using the following equation (Equation (4)) [52]:(4)$$P_{c}=\frac{2D}{\left(1-p\right)}$$
where *D* is the mean fiber diameter, *p* is approximately taken to represent overall porosity, and (1 − *p*) is the average total projected area of fibers per unit area.

#### 3.3.2. In Vitro Drug Release Tests and Release Kinetics

The in vitro dissolution of Cur-loaded nonwovens of 5–10 mg was examined in 10 mL release media, including phosphate-buffered saline (PBS, pH 7.4) with 0.1% Tween 80 to support the low solubility of Cur in aqueous media and 0.1% sodium azide to prevent bacterial growth at 37 °C, and shaken at 70 rpm. At predetermined time intervals for 14 days, 500  μL aliquots were withdrawn for sampling and replaced with an equal volume of release media to maintain a constant volume. The release of Cur from the samples was monitored by a UV–visible microplate spectrophotometer (Multiskan Go, Thermo Fisher Scientific, Waltham, MA, USA) at a wavelength of 428 nm.

The release profile of each sample was performed in five repetitions. The cumulative percentage of Cur released was determined using a predetermined standard calibration curve of Cur in the same solvent system. The Cur release rate was additionally determined as a derivative of the cumulative release profile over time.

The release data obtained were fitted to the Gallagher–Corrigan model [35,56] (Equation (5)) by using Python (Phyton 3.12.12), to determine the release kinetics and mechanism of Cur from fibrous membranes. This model was chosen due to its ability to describe the systems where multiple drug release mechanisms are involved. Given the complexity of release behavior observed, such a comprehensive model was considered to fit better than simpler models, which are limited to specific release kinetics or geometries.(5)$$f_{t}=f_{B}(1-e^{-k_{1}t})+(1-f_{B})\;[\frac{e^{k_{2}t-k_{2}t_{m}}}{1+e^{k_{2}t-k_{2}t_{m}}}]$$
where $f_{t}$ is the fraction of the drug released at time *t*; $f_{B}$ is the fraction of the drug released at the bursting stage; $k_{1}$ is the release constant in the first stage; $t_{m}$ is the maximum release time; and $k_{2}$ is the release constant during the second stage of release.

#### 3.3.3. Surface Wettability Test

The water contact angle (WCA) was measured to assess the hydrophilicity of the nanofiber membranes. The measurement was performed by the sessile drop method (OCA 15Pro, Dataphysics, Filderstadt, Germany). Considering the hydrophilicity of the material, the WCA values were recorded immediately after the deionized water (2 microliters) dropped freely onto the surface of the square nanofiber membrane.

#### 3.3.4. Cytotoxicity Evaluation 

Cytotoxicity tests were conducted using the L929 line of fibroblasts (Sigma Aldrich, St. Louis, MO, USA). Cells were cultured in a 25 cm^2^ flask containing High Glucose Dulbecco’s Modified Eagle’s Medium (DMEM) (Life Technologies, Waltham, MA, USA), supplemented with 10% fetal bovine serum (FBS) (Life Technologies, Waltham, MA, USA) and 1% antibiotics (Life Technologies, Waltham, MA, USA). The cells were incubated at 37 °C in a 5% CO_2_ environment. To detach the cells from the flask, they were washed with PBS and then treated with 2 mL of a 0.05% trypsin solution, followed by incubation for a few minutes. Once the cells were harvested, 10 mL of culture medium was added, and the mixture was centrifuged at ambient temperature. The resulting pellet was resuspended in the culture medium to achieve the desired cell density. Various studies were performed to evaluate cellular cytotoxicity to 2D membranes and 3D scaffolds, including analysis of viability on extracts and cellular morphology.

##### Cytotoxicity Assay

Extracts for in vitro tests were prepared by placing 8 mg of each sample type into a 24-well plate, as previously described [44]. Samples were immersed in 2 mL of culture medium per well, maintained at 37 °C, and gently stirred for 24 h. For reference, wells with and without samples were filled with the medium. Simultaneously, an L929 cell suspension was seeded into another 48-well plate at a density of 1.5 × 10^4^ cells/well and incubated for 24 h. The culture medium in cell-seeded wells was then replaced with 100% extracts of the analyzed samples. The plate was incubated for another 24 h. Subsequently, extracts were removed, and each well was filled with 180 μL of PBS and 20 μL of Presto Blue reagent. The plate was incubated for an additional 60 min. Finally, 100 μL from each well was transferred to a 96-well plate, and fluorescence was measured using 530/620 nm excitation/emission filters on a Fluorescent Accent FL Thermo Fisher Scientific instrument (Waltham, MA, USA). Results were correlated with the control (Tissue Culture Plate TCP), which exhibited 100% viability.

##### Fibroblast Morphology

Cellular morphology after exposure to extracts was verified using fluorescence microscopy. L929 cells were seeded at a density of 20 × 10^4^/well in 400 μL of medium in a 48-well plate. After 24 h, the medium was replaced with extracts. Following another 24 h of cultivation, cells were fixed in 3% formaldehyde for 20 min and then treated with 0.01% Triton X-100 for 5 min to permeabilize cell membranes. Cellular nuclei and cytoskeletons were stained for 30 min with a solution of ActinGreen and NucBlue, which bind to the actin skeleton and nuclear DNA, respectively. Images were captured using a Leica AM TIRF MC microscope at 100× and 400× magnifications.

#### 3.3.5. Statistical Analysis

Statistical evaluation of cytotoxicity was performed using GraphPad Prism 8.0.1 Software, with significance set at *p* < 0.05. Two-way ANOVA followed by Tukey’s multiple comparisons test was used where appropriate. The *p* below 0.05 was considered to be statistically significant, where 0.05 > *p* > 0.01 is assigned as “*”, 0.01 > *p* > 0.001 is set as “**”, while *p* < 0.001 is assigned as “***”.

## 4. Conclusions

In the present study, core–shell fiber meshes with the addition of curcumin in different compositions were developed by coaxial and triaxial electrospinning. The fibrous meshes were characterized and evaluated along with their Cur release profiles with the goal of better utilizing the promising triaxial core–shell fibers for drug delivery in tissue engineering applications.

The successful electrospinning process led to the fabrication of homogeneous and bead-free fibers, particularly for the composition with PCL as a core and PLGA as a shell. Additionally, this composition indicates more evenly distributed layer thicknesses across the samples. The multilayered structure was confirmed by FM with the localization of RhB in the targeted middle layer of the triaxial fibers in both compositions. According to the pore size analysis, composition with PCL as a core and PLGA as a shell provides a better balance between being cell occlusive and the permeability of essential factors, while both compositions have pore sizes small enough to prevent cell migration. 

The in vitro Cur release studies indicated that coaxial and triaxial systems of both compositions exhibited a biphasic release profile, composed of an initial burst release, followed by a sustained release. Triaxial fiber architecture provided an essential decrease in the initial burst release and led to a more controlled and extended release over time. This behavior of regulating release for a longer time was more pronounced with PCL as a core and PLGA as a shell layer. The more effective diffusion barrier formed by PLGA can be explained by the glassy nature of PLGA at 37 °C, contrary to the rubbery state of PCL, being well above Tg. Such variations demonstrated the importance of shell layer characteristics, indicating that the composition with PCL as a core layer and PLGA as a shell layer is preferred for sustainable drug release. Furthermore, higher curcumin loading (3%) resulted in an increased release, suggesting that the release mechanism is driven by the diffusion gradient. This is also confirmed by fitting the experimental data to the Gallagher–Corrigan mathematical model.

In addition to these, cytotoxicity analysis demonstrated that all fibrous meshes exhibited no toxic effect on fibroblast cells. Furthermore, the increment of Cur concentration in triaxial fibers did not significantly affect the cellular viability, while the coaxial fibers showed decreasing cellular viability. This result was another confirmation of the shell layer successfully acting as a barrier layer.

While this work provides important insights into the design–performance relationship of multicomponent fibers, further structural and physicochemical analyses, as planned in a complementary study, will deepen the understanding of the underlying mechanisms and provide a foundation for further optimization.

## Figures and Tables

**Figure 1 molecules-30-04241-f001:**
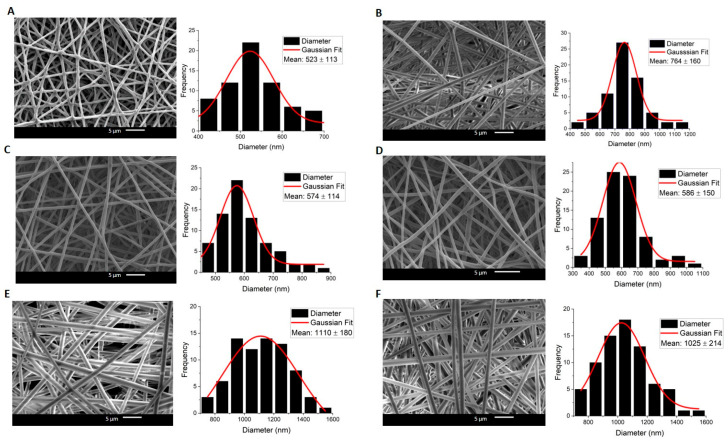
SEM images and Gaussian approximation of fiber diameter distributions for Composition 1: (**A**) Co_Plain, (**B**) Tri_Plain, (**C**) Co_1%, (**D**) Co_3%, (**E**) Tri_1%, (**F**) Tri_3%.

**Figure 2 molecules-30-04241-f002:**
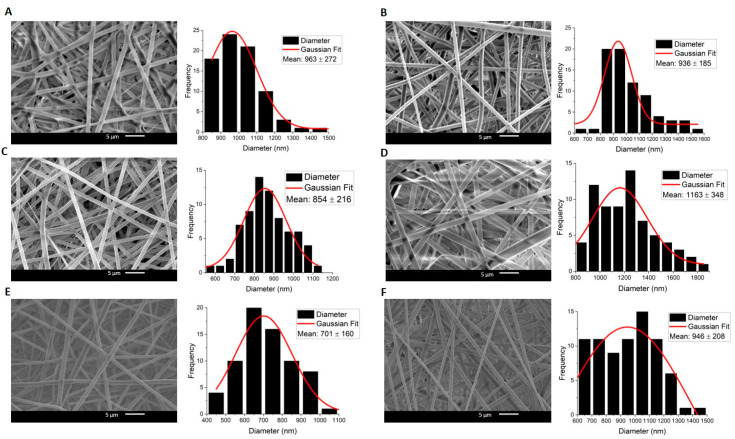
SEM images and Gaussian approximation of fiber diameter distributions of Composition 2: (**A**) Co_Plain, (**B**) Tri_Plain, (**C**) Co_1%, (**D**) Co_3%, (**E**) Tri_1%, (**F**) Tri_3%.

**Figure 3 molecules-30-04241-f003:**
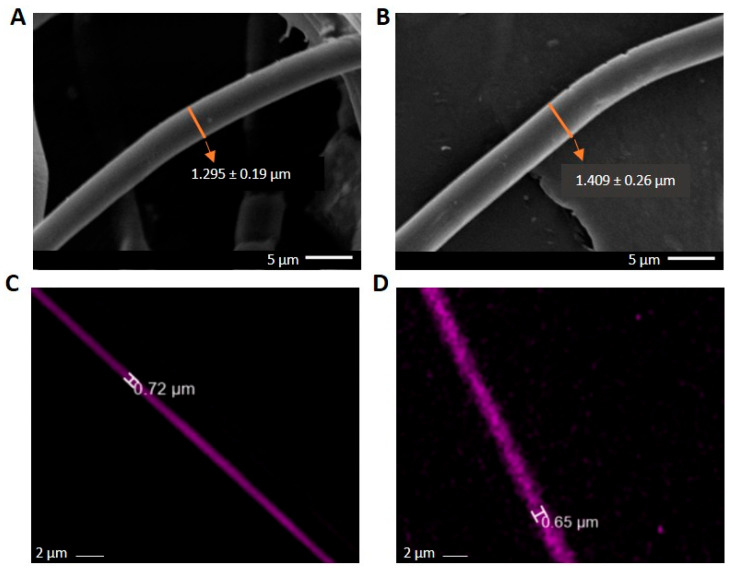
RhB-loaded triaxial fibers. (**A**,**B**) SEM images of triaxial fibers of Composition 1 and Composition 2, respectively. (**C**,**D**) FM images of RhB-loaded triaxial fibers of Composition 1 and Composition 2, respectively.

**Figure 4 molecules-30-04241-f004:**
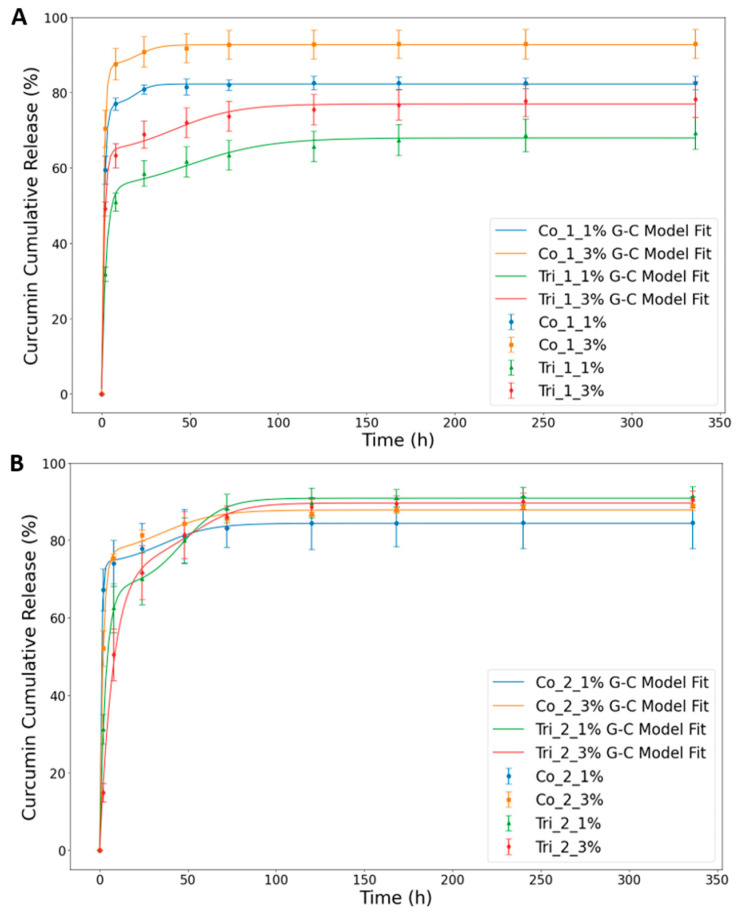
Experimental data of the cumulative release of curcumin and the fitted curve obtained by applying the Gallagher–Corrigan (G-C) model: (**A**) Composition 1, (**B**) Composition 2. The suffixes “_1” and “_2” refer to Composition 1 and Composition 2, respectively.

**Figure 5 molecules-30-04241-f005:**
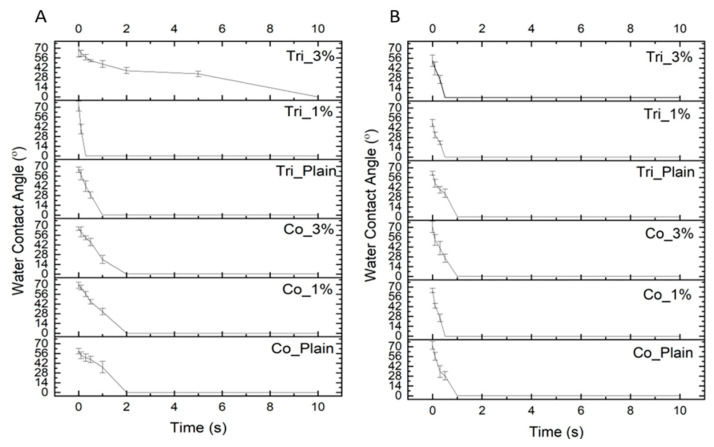
Water contact angle of fibers vs. time of contact: (**A**) Composition 1, and (**B**) Composition 2.

**Figure 6 molecules-30-04241-f006:**
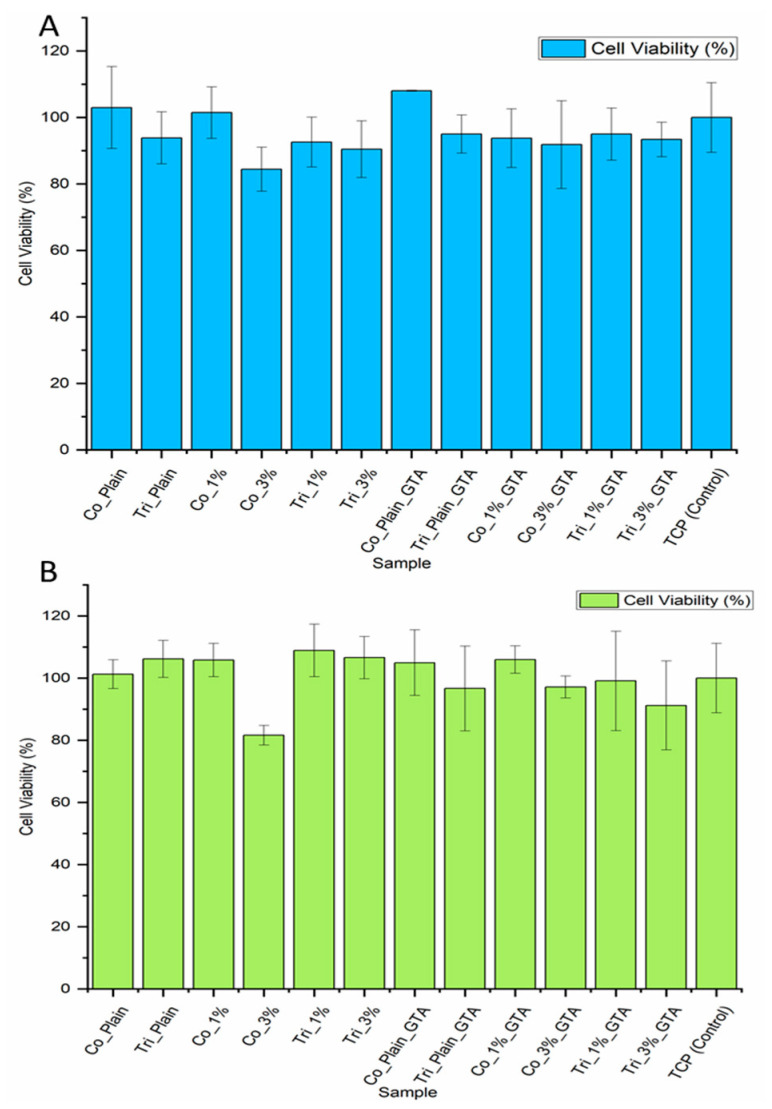
Cell viability of L929 cells after exposure to testing groups for 72 h: (**A**) Composition 1, (**B**) Composition 2; Co: Coaxial, Tri: Triaxial, GTA: Glutaraldehyde vapor crosslinking. (Mean ± SD, *n* = 3).

**Figure 7 molecules-30-04241-f007:**
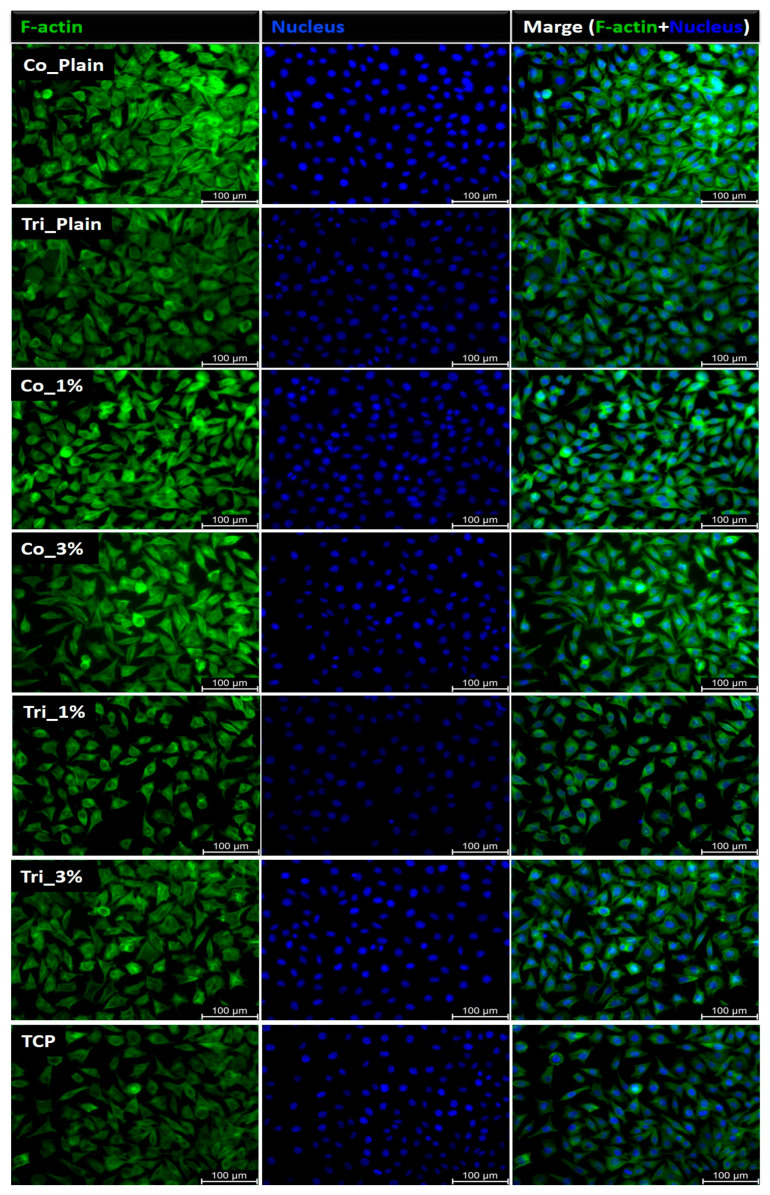
Fibroblast morphology of samples of Composition 1: L929 cells cultured for 24 h in contact with sample extracts in comparison to TCP as a control; actin skeleton in green (left column), cellular nucleus in blue (middle column), merged nucleus, and actin images (right column).

**Figure 8 molecules-30-04241-f008:**
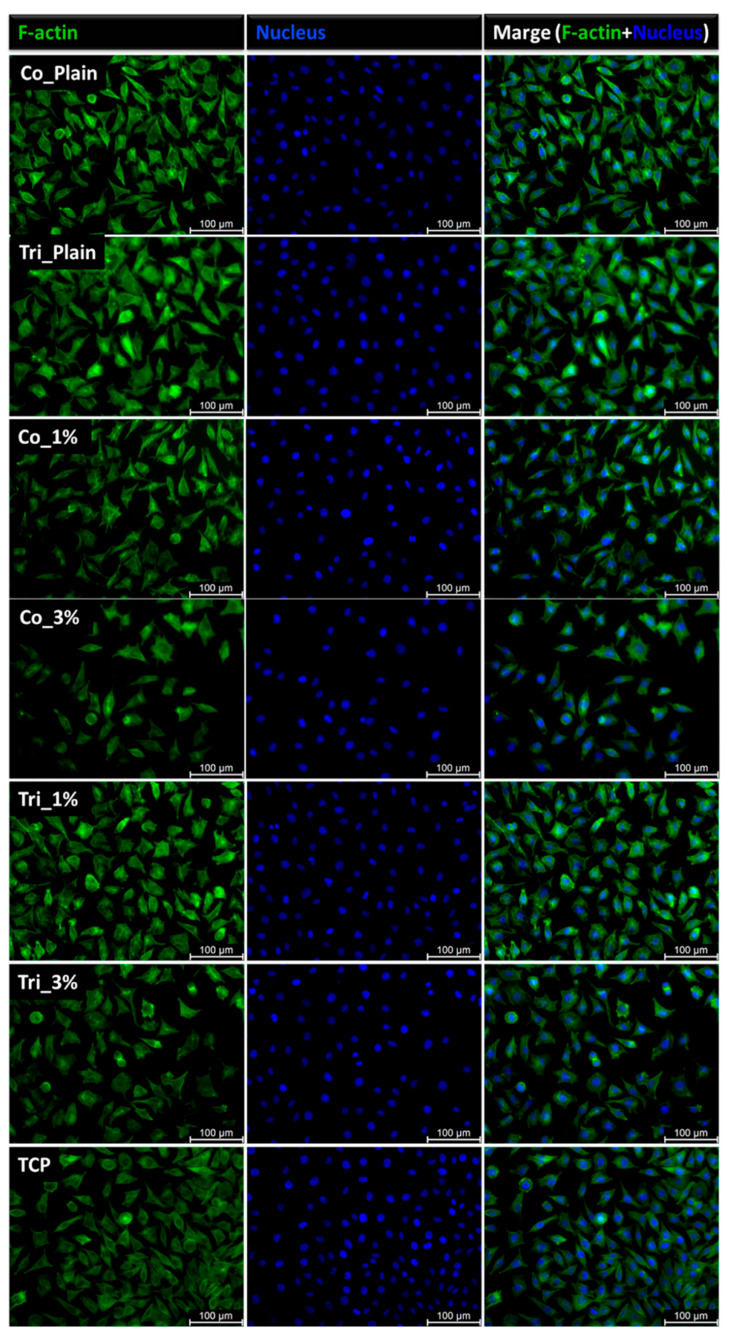
Fibroblast morphology of samples of Composition 2: L929 cells cultured for 24 h in contact with sample extracts in comparison to TCP as a control; actin skeleton in green (left column), cellular nucleus in blue (middle column), merged nucleus, and actin images (right column).

**Figure 9 molecules-30-04241-f009:**
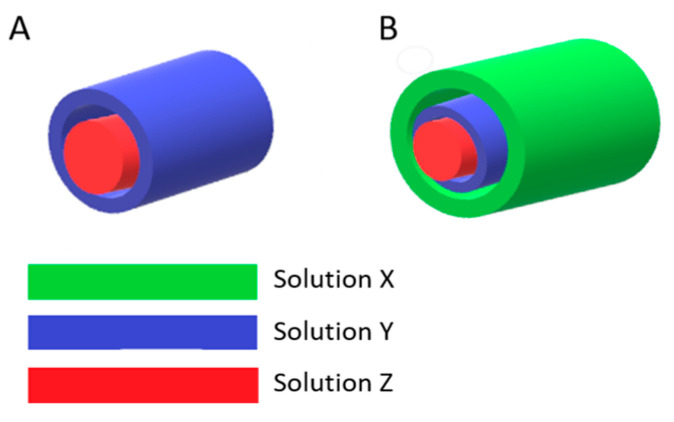
Illustration of the architecture of fibers developed: (**A**) coaxial fiber; (**B**) triaxial fiber.

**Figure 10 molecules-30-04241-f010:**
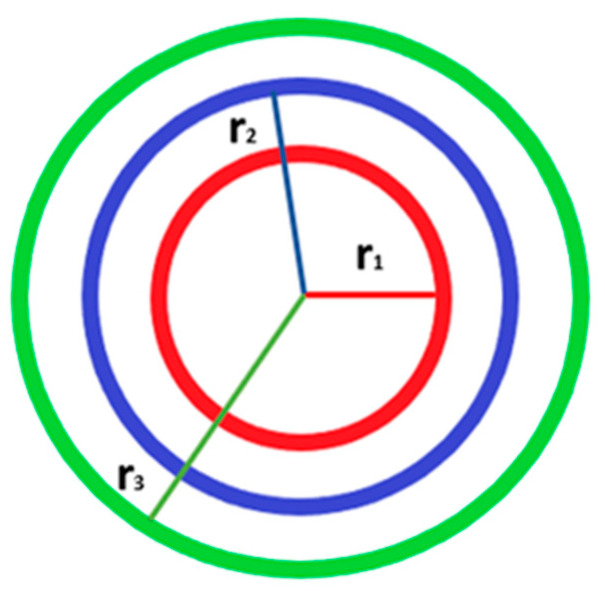
Representative visualization of the core–shell fiber for theoretical calculations; radius of the layers known from the triaxial needle’s product specifications; r_1_, r_2_, r_3_ are the radius of the core, middle, and shell layers, respectively.

**Table 1 molecules-30-04241-t001:** Fibers’ layer thicknesses for Compositions 1 and 2. Prefixes C1 and C2 represent Compositions 1 and 2, respectively.

Sample	C1. Core (nm)	C1. Middle (nm)	C1. Shell (nm)	C2. Core (nm)	C2. Middle (nm)	C2. Shell (nm)
Co_Plain	238.26	-	284.73	438.72	-	524.27
Tri_Plain	191.62	228.99	343.37	187.81	224.44	523.74
Co_1%	261.50	-	312.49	389.06	-	464.93
Co_3%	266.96	-	319.03	529.83	-	633.16
Tri_1%	278.42	332.70	498.87	140.66	168.09	392.24
Tri_3%	257.10	307.23	460.67	189.82	226.84	529.33

**Table 2 molecules-30-04241-t002:** Porosity and average pore size of electrospun fiber meshes. Prefixes C1 and C2 represent Compositions 1 and 2, respectively.

Sample	C1. Porosity (%)	C1. Pore Size (µm)	C2. Porosity (%)	C2. Pore Size (µm)
Co_Plain	76 ± 3	4.5 ± 0.5	82 ± 1	11 ± 0.6
Tri_Plain	79 ± 6	7.8 ± 2.4	68 ± 8	6.1 ± 1.5
Co_1%	80 ± 1.6	5.8 ± 4.5	76 ± 8	8 ± 3.5
Co_3%	80 ± 6	6.4 ± 2	71 ± 3.5	8.2 ± 1
Tri_1%	78 ± 2.5	10.3 ± 1.1	66 ± 2	4.2 ± 2.7
Tri_3%	81 ± 1	11 ± 0.6	63 ± 5	5.2 ± 0.7

**Table 3 molecules-30-04241-t003:** Results obtained for curcumin release using the Gallagher–Corrigan model.

Samples	f_B_	k_1_	k_2_	t_m_	R^2^
Co_1_1%	77	0.739	0.225	19.11	0.9999
Co_1_3%	87	0.818	0.164	20.27	0.9999
Tri_1_1%	53	0.396	0.043	49.10	0.9961
Tri_1_3%	63	0.696	0.055	40.79	0.9975
Co_2_1%	73	1.159	0.072	35.10	0.9996
Co_2_3%	76	0.550	0.065	35.05	0.9987
Tri_2_1%	68	0.299	0.083	46.98	0.9997
Tri_2_3%	75	0.130	0.075	53.13	0.9986

The suffixes “_1” and “_2” refer to Composition 1 and Composition 2, respectively. **f_B_**: fraction of drug released at the bursting stage; **k_1_**: release constant in the first stage; **t_m_**: maximum release time; and **k_2_**: release constant during the second stage of release.

**Table 4 molecules-30-04241-t004:** Constituent parts of fibers in Composition 1.

Sample	Core Layer	Middle Layer	Shell Layer
Co_Plain	PCL	GT	-
Tri_Plain	PCL	GT	PLGA
Co_1%	PCL	GT with 1% Cur (wt.)	-
Co_3%	PCL	GT with 3% Cur (wt.)	-
Tri_1%	PCL	GT with 1% Cur (wt.)	PLGA
Tri_3%	PCL	GT with 3% Cur (wt.)	PLGA

**Table 5 molecules-30-04241-t005:** Constituent parts of fibers in Composition 2.

Sample	Core Layer	Middle Layer	Shell Layer
Co_Plain	PLGA	GT	-
Tri_Plain	PLGA	GT	PCL
Co_1%	PLGA	GT with 1% Cur (wt.)	-
Co_3%	PLGA	GT with 3% Cur (wt.)	-
Tri_1%	PLGA	GT with 1% Cur (wt.)	PCL
Tri_3%	PLGA	GT with 3% Cur (wt.)	PCL

## Data Availability

Raw data will be made available by the corresponding author upon reasonable request.

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
