# Peer review of "Triaxial Electrospun Nanofiber Membranes for Prolonged Curcumin Release in Dental Applications: Drug Release and Biological Properties"

_molecules, 2025, doi:10.3390/molecules30214241_

Round 1

Reviewer 1 Report

Comments and Suggestions for Authors

The manuscript presents an experimental study on the fabrication and evaluation of triaxial electrospun fiber membranes composed of polycaprolactone, PLGA and gelatin as delivery systems for curcumin. The work demonstrates that the arrangement of polymer layers (core–shell positioning) affects drug release kinetics and cytocompatibility. Overall, the study is well-structured and methodologically sound; however, the scientific novelty is incremental, and several aspects of presentation and contextual framing require improvement to strengthen the manuscript’s impact and clarity.

Comments and recommendations:

- The manuscript explores electrospinning (a known technique in drug delivery) using well-established materials (PCL, PLGA, and gelatin) and curcumin as a model compound. While the comparison between coaxial and triaxial architectures and the inversion of core-shell polymer layers (Composition 1 vs. Composition 2) provide useful insights into design -performance relationships, the study’s innovation lies mainly in design optimization, not in the introduction of new materials, mechanisms, or applications. The introduction section should more clearly state why this particular structural comparison is important and how it advances existing knowledge. The rationale and expected benefits of rearranging the polymer layers should be emphasized as the main source of novelty. 

- The abstract begins with technical details and lacks a clear statement of purpose, motivation, and significance. Quantitative results are missing, making the findings appear more qualitative and descriptive.

- The terms "Composition 1" and "Composition 2" are used frequently, but their definitions appear late in the manuscript. For reader clarity, both should be clearly described in the early part of the Results and Discussion section, not only in the experimental section.

- Table 1 reports “thicknesses” (presumably average fiber diameters) but lacks units.

- The flow of information between methods, results, and discussion could be improved for coherence. Some results are discussed before sufficient methodological detail is provided.

- The manuscript includes valuable morphological and release studies but fails key physicochemical and mechanical characterizations that are critical for tissue engineering applications. No data are provided on mechanical strength, flexibility, or degradation rate, which are essential to assess the suitability of the membranes for biomedical use. Include or at least discuss potential analyses.

- The conclusions are detailed but somewhat repetitive. They should be made more concise and focus on clear cause - effect relationships (for example, explaining how the PLGA shell specifically influences diffusion....). The authors should also link their findings to practical design principles that could guide the development of future drug delivery systems.

Reviewer 2 Report

Comments and Suggestions for Authors

This study focuses on triaxial electrospun nanofiber membranes for prolonged curcumin release in dental applications. It centers on the preparation of drug carriers using triaxial electrospinning technology, optimizing curcumin release performance to meet the needs of dental tissue engineering. By closely integrating material design with medical applications, the research outcomes demonstrate clear potential for clinical translation. The study design is rational and practically valuable; however, there is room for improvement in experimental details, data presentation, and discussion depth. Overall, further refinement is required to enhance the rigor and persuasiveness of the research.

  1. Incomplete chart information:

Figures 1 and 2 only present SEM images and Gaussian fitting of fiber diameter distributions, but fail to label the specific range of fiber diameter values and standard deviations in the figures. The headers of Tables 1 and 2 are confusing and need to be reorganized to clearly indicate the data attribution corresponding to Composition 1 and Composition 2, thereby avoiding ambiguity.

  1. Inconsistent reference formatting:

Some references have formatting errors in author names, journal titles, volume-issue numbers, and page numbers (e.g., the abbreviation "J. Cell. Physiol." in Reference 15 is non-standard). All references should be uniformly revised in accordance with the reference formatting requirements of the Molecules journal to ensure the accuracy of citation formatting.

  1. Lack of direct characterization data for polymer glass transition temperature (Tg):

In the discussion, differences in drug release behavior are attributed to the Tg discrepancy between PCL and PLGA. However, the actual Tg values of the two polymers and the fiber membranes were not determined via differential scanning calorimetry (DSC) or dynamic mechanical analysis (DMA). Reliance solely on literature-reported data fails to rule out the impact of the electrospinning process on polymer Tg. Relevant characterization results and data need to be supplemented.

  1. Unvalidated porosity and pore size measurement methods:

Formulas (eq.3, eq.4) were used to calculate porosity and average pore size, but the applicability of these formulas to triaxial electrospun nanofiber membranes was not explained. Additionally, no comparison was made with measurement results from classical methods such as mercury intrusion porosimetry or nitrogen adsorption-desorption. This makes it impossible to confirm the reliability of the calculated data. Method validation data should be supplemented, or literature citations should be provided to explain the basis for the applicability of the formulas.

  1. Missing quantitative analysis of cell morphology:

In the biocompatibility evaluation, only the fact that cell viability exceeded 80% was mentioned. However, quantitative statistics on indicators such as the spreading area, filopodia length, and nuclear morphology of L929 fibroblasts were not conducted, nor was the impact of different samples on the arrangement of the cytoskeleton compared. This makes it impossible to fully demonstrate the supportive effect of the materials on cell growth. Quantitative analysis data of fluorescence images (e.g., analysis of cell spreading area using ImageJ software) need to be supplemented.

  1. Some relevant papers can be cited:

https://doi.org/10.3390/BIOM12060794

Comments on the Quality of English Language The English in the paper demonstrates a good command of vocabulary and grammar, with clear and coherent expressions that effectively convey the author's viewpoints.

Reviewer 3 Report

Comments and Suggestions for Authors

- The abstract needs to be summarized, because in the present form has a lot of information
- Line 93 “Cur-loaded core-shell fibers were optimized.” Improve this sentence; “optimized” was very general
- Lines 100-109, the manuscript mentioned “Composition 1 and Composition 2”, it was difficult to understand. Try to change compositions 1 and 2 for experimental conditions
- Figures 1 and 2 were confused, because it has two figures A, B, C, etc. Magnification was difficult to see (right figures), and it was not possible to see the text information in the left figures. Please improve it
- The manuscript has some typos, revise carefully and correct them
- Lines 133-134 “diameters obtained by SEM” explain the procedure to obtain diameters by SEM
- Figure 6, explain why cell viability showed more than 100%
- 3.1. Materials, including information about purification materials or the sentence “used as received”
- The manuscript has some interesting results, but doesn’t have a discussion, including it for all figures and tables
- Why is it important to include figure 10
- Try to include more characterization methods, maybe mechanical properties, thermal analysis, RMN or FT-IR, etc.
- Line 166 “The cumulative Cur release profiles, as measured by a UV spectrometer (Fig. 6)” but figure 6 has this caption “Figure 6.Cell viability of L929 cells after exposure to testing groups for 72 h.A: Composition 1, B: Composition 2,C: Coaxial, T: Triaxial, GTA: Glutaraldehyde vapor crosslinking. (Mean ±SD, n=3)”
- The manuscript must present figures about the load and drug release 

Reviewer 4 Report

Comments and Suggestions for Authors
  1. The Gaussian approximation of fiber diameter distributions is not clear at Figure 1.
  2. The Gaussian approximation of fiber diameter distributions is not clear at Figure 2.
  3. The porosity appears between 60% and 80% in Table 2. Traditionally, porosity should be greater than 80%. Therefore, this study raises doubts about its applicability for subsequent research.
  4. Why is there no error bar for CO_Plain_GTA in Figure 6 A?
  5. “The process was carried out at the temperature range 365 of 22− 25 â—¦C and humidity range of 30-40 %. Other process parameters were as follows: 366 distance between needle and collector 15 cm, voltage applied to the needle 12-15 kV”. The authors must explain the choose the process parameters and values for electrospun process.
  6. The CO2 locates at line 493, page 18. CO2 must modify to CO2.
Comments on the Quality of English Language

The English writing must be improved.

Round 2

Reviewer 1 Report

Comments and Suggestions for Authors

The authors have addressed all comments and made substantial improvements to the manuscript. In my view, the revised version is suitable for publication in its present form.

Reviewer 4 Report

Comments and Suggestions for Authors

The English writing can be improved.

Comments on the Quality of English Language

The English writing can be improved.